# "…*I feel like I am just staying here waiting for death*": A qualitative study of the lived experiences of people with advanced illness in refugee settlements in Uganda

Ning Song[1], Agatha Aduro-Agema[1], Eve Namisango[2], Sahla Aroussi[3], Ping Guo[4], Mhoira Leng[5], Bassey Ebenso[1], Shaunna Burke[6], Vicky Opia[7], Raphael Rabonye[8], Emmanuel Mutoto[8], Felix Muehlensiepen[9,10], Hanna Kaade[9], William E. Rosa[11], Eme Asuquo[1], Richard A. Powell[12], Matthew J. Allsop[1]*

**1** Leeds Institute of Health Sciences, University of Leeds, Leeds, United Kingdom, **2** African Palliative Care Association, Kampala, Uganda, **3** School of Politics and International Studies, University of Leeds, Leeds, United Kingdom, **4** Department of Nursing and Midwifery, University of Birmingham, Birmingham, United Kingdom, **5** Palliative Care in Humanitarian Aid Situations and Emergencies (PallCHASE), Edinburgh, United Kingdom, **6** School of Biomedical Sciences, University of Leeds, Leeds, United Kingdom, **7** Peace Hospice, Adjumani, Uganda, **8** International Rescue Committee, Kampala, Uganda, **9** Center for Health Services Research, Brandenburg Medical School Theodor Fontane, Rüdersdorf, Berlin, Germany, **10** Department of Cardiology, Angiology and Intensive Care Medicine, Deutsches Herzzentrum der Charité, Berlin, Germany, **11** Department of Psychiatry and Behavioral Sciences, Memorial Sloan Kettering Cancer Center, New York, New York, United States of America, **12** Imperial College London, London, United Kingdom

* m.j.allsop@leeds.ac.uk

## Abstract

Refugees and migrants with advanced illness frequently face fragmented and inconsistent access to health services. Structural barriers delay symptom management, undermine continuity of care, and limit access to palliative support. A clearer understanding of their lived experiences is essential to inform equitable health system responses and inclusive models of care. This study explored the lived experiences of individuals with advanced illness in three refugee settlements in Uganda to identify contextually grounded strategies for improving access to and delivery of palliative care. In-depth semi-structured interviews were conducted with 44 purposively sampled refugees living with advanced illnesses, reflecting varied conditions, ages, and care experiences. A descriptive phenomenological approach combined with reflexive thematic analysis was used to identify key patterns and meanings related to illness experiences and care access. Participants' pre-displacement contexts—often characterised by traumatic events—intensified current physical and psychological symptoms. A recurring theme was the loss of ability to care for their families due to declining health and limited functional capacity within the settlement. Unmet basic needs and restricted access to health services further hindered effective illness management. Displacement, trauma, and scarce resources collectively undermined health and wellbeing. Displaced individuals with advanced illness in Uganda face

**Data availability statement:** Due to the sensitive nature of the qualitative data and the heightened risk of re-identification within refugee settlement contexts, the full interview transcripts cannot be publicly shared. This restriction is mandated by the protocol for which ethical approval was granted. Researchers may request controlled access to de-identified transcripts from the University of Leeds Research Data Management team (researchdataenquiries@leeds.ac.uk), subject to a data access agreement and, where necessary, additional ethics approval.

**Funding:** This research was undertaken with support from the UK Medical Research Council Impact Acceleration Fund (MR/X502789/1MRC). MA is funded through a Yorkshire Cancer Research University Academic Fellowship. WER is partially supported by NIH/NCI comprehensive cancer center award P30 CA008748 and the Robert Wood Johnson Foundation Harold Amos Medical Faculty Development Program. RAP was supported by the National Institute for Health and Care Research (NIHR) Applied Research Collaboration Northwest London. The views expressed by RAP are not necessarily those of the NIHR or the Department of Health and Social Care, London, UK. No funders had any role in the study design, data collection and analysis, decision to publish, or preparation of the manuscript.

**Competing interests:** The authors have declared that no competing interests exist.

severe and interconnected challenges, including inconsistent access to healthcare, unmet psychosocial and spiritual needs, and limited support from non-governmental organisations (NGOs) and palliative care services. Culturally responsive, community-engaged strategies coordinated across humanitarian and national systems are urgently needed to address the multifaceted needs of this population in a coordinated and sustainable way.

## Notice

In keeping with best practices for trauma-informed scholarship, we preface this work with a clear notice that the following pages address gendered harms, including sexual violence, within refugee and humanitarian crisis settings. Because many palliative-care needs—pain control, psychosocial support, end-of-life decision-making—are profoundly shaped by prior trauma, the study includes first-person accounts of sexual assault, violence within conflict, and coercive harms. While such evidence is essential for designing compassionate, population-appropriate palliative care services, portions of the text may trigger distress. Readers who prefer to avoid such content may wish to review only the Abstract and Conclusions sections, where graphic detail is omitted. We present these data with utmost respect for participants, in line with trauma-informed research standards and humanitarian ethics. The goal is to ensure that evidence—however challenging and unsettling—guides equitable palliative care delivery that more precisely meets the needs of people living with advanced illness in displacement settings.

## Introduction

Global displacement has reached unprecedented levels, driven by conflict, political instability, and environmental crises [1]. Most displaced people resettle in low- and middle-income countries (LMICs) where already strained health systems must meet the needs of both host and refugee populations [2]. While humanitarian responses traditionally focus on acute and infectious conditions, there is growing recognition that many displaced individuals live with chronic, progressive, and life-limiting illnesses requiring long-term care and support [3]. Conditions such as advanced cancer, end-stage organ failure, and neurodegenerative diseases often go undiagnosed or untreated in displacement settings, resulting in inadequate pain control and minimal psychosocial or caregiver support [4,5]. These inequities can be understood through the lens of structural violence, which highlights how political and social systems systematically expose marginalised populations to preventable suffering [6]. The interaction of displacement with chronic illness also reflects a syndemic dynamic, where multiple social and biological risk factors co-occur and reinforce one another under conditions of inequality [7].

Palliative care is "…the active holistic care of individuals across all ages with serious health-related suffering due to severe illness and especially of those near the end of life. It aims to improve the quality of life of patients, their families and their caregivers" [8]. Despite global advocacy for palliative care integration into universal health coverage and recognition of palliative care as an ethical obligation of all health systems [9], it remains marginal in humanitarian health strategies [10,11]. There is limited empirical research on the palliative care needs of displaced populations in LMICs [12]. However, there are known critical gaps in palliative care provision that exist in crisis contexts, including inconsistent pain management and a lack of context-sensitive clinical and ethical guidance [13] and limited recognition of the importance of alleviating suffering and ensuring a dignified death [14]. For providers of palliative care in crisis settings, too, moral distress and systemic constraints in delivering end-of-life care are commonly experienced [15].

Across Africa, migration involves large numbers of international migrants, with most movement within the region [16]. Uganda hosts the largest refugee population in Africa, with more than 1.7 million refugees, primarily from South Sudan and the Democratic Republic of Congo [2]. Across Uganda, significant progress has been made in integrating palliative care into its health system, with preliminary integration into mainstream service provision [17]. Uganda has served as a leading example of palliative care development in Africa, supported by the incorporation of palliative care into education, comprehensive training of healthcare professionals in opioid treatment, and the authorisation of specially trained palliative care nurses to prescribe opioids [18]. However, refugees reside mainly in settlements where healthcare services are fragmented, essential medications are scarce, and specialist care is largely unavailable. Cultural and linguistic barriers further hinder access to services and effective communication [19]. These intersecting conditions—displacement, poverty, weak infrastructure, and serious illness—can create a syndemic environment in which prolonged and preventable suffering becomes normalised. Despite Uganda's advances in national palliative care integration, there is little empirical understanding of how displaced populations with advanced illness experience care and support in refugee settlements. As palliative care provision within Uganda's national health system advances, service organisations in refugee settlements remain distributed across humanitarian, governmental, and community actors. This study explores the lived experiences of individuals with advanced illness in refugee settlements in Uganda, addressing a critical gap in the evidence base on palliative care in humanitarian contexts. By centring their perspectives, the study aims to convey experiences and consider how to develop responsive and culturally grounded palliative care in humanitarian contexts.

## Materials and methods

### Ethics statement

The study was approved by the Hospice Africa Uganda Research Ethics Committee (reference HAU-2021–4). All study participants received study information details and provided their written informed consent before voluntary participation.

### Methodological approach

We framed this research within the phenomenological paradigm [20], enabling the team to explore the meanings, values, feelings, and concerns of individuals with advanced illness in the context of refugee settlements in Uganda. We combined a descriptive phenomenological approach with a reflexive thematic analysis. The study forms part of a wider ongoing partnership between the African Palliative Care Association, University of Leeds, and clinical and academic partners across Africa to develop and evaluate new models of palliative care delivery in the region. The Consolidated Criteria for Reporting Qualitative Research (COREQ) checklist [21] was used to guide the reporting of the study.

### Setting

Uganda hosts a refugee population chiefly from South Sudan and the Democratic Republic of the Congo (DRC) across 13 major rural settlements [2]. Data for this project were collected across three settlements: Nyumanzi Refugee Settlement,

Kyaka II Refugee Settlement, and Bidibidi Refugee Settlement. The three settlements reflect a spectrum of settlements in Uganda: Nyumanzi Refugee Settlement, a compact, border-adjacent entry point in north-west Uganda; Kyaka II Refugee Settlement in the west supports Congolese inflows; and Bidibidi Refugee Settlement, now Uganda's largest settlement, which is similar to a small city. Table 1 provides summary statistics regarding the size and composition of the populations in the three settlements.

Across the three settlements, care for chronic and advanced illness is delivered through a combination of settlement-level primary health facilities, district or regional referral hospitals, and periodic outreach from hospice or hospital-based palliative care teams. Primary facilities provide holistic care, including symptom management, but have limited diagnostic capacity and intermittent availability of essential medicines, including opioids. Referral pathways to district hospitals exist for specialist review or investigations; however, these can involve long travel distances, administrative delays, and inconsistent communication regarding follow-up. Palliative care teams visited settlements to provide counselling, oral morphine where available, and home-based support; however, the regularity of visits varied between sites due to staffing, caseload, and logistical constraints.

## Sample

A standardised approach to identification and recruitment was applied across all three sites. Initial clearance was sought and granted by the Office of the Prime Minister, followed by the United Nations High Commissioner for Refugees (UNHCR), and then subsequently by the district health officers and regional district officers for the regions in which the settlements were located. The research team then worked with the regional commanders (who oversee operations, personnel, and infrastructure within the region in which settlements are located) and the health service providers to determine suitable sites for recruiting participants. Health service providers generated lists of patients registered to receive palliative care. The list was reviewed by community health service coordinators and community health workers, who conducted rounds in the community across settlements to determine which palliative care patients were alive, had migrated, or had died.

Potential participants were approached by community health workers who outlined details of the study. As part of the consent process, potential participants were informed that they were free to choose whether to participate and could opt out at any time, and they could do this by informing a community health worker or the person conducting the interview. Participants were informed that their decision to participate or withdraw would not affect the type or quality of care they were receiving. Participants could choose to have the interview at their home or a nearby health centre. All participants received a transport refund if they attended a health centre for the interview. All participants received refreshments and snacks during interviews. Participants were eligible for this study if they were 18 years or older, had been diagnosed with advanced disease, could communicate fluently with health workers (interviewers) who were multilingual and conversant in

**Table 1. A summary of the participating settlements, population size and composition.**

| Category | Nyumanzi | Kyaka II | Bidibidi |
| --- | --- | --- | --- |
| Year opened | 2014 | 2003 | 2016 |
| Refugees (2024–25) | ~47 k | >113 k | ~270 k |
| Host-district pop. (2024 census) | 300 k | 501 k | 945 k |
| Footprint (km²) | ~67 | 81.5 | 250 |
| Main countries of origin | South Sudan | DRC, Rwanda | South Sudan (small DRC cohort) |
| Geographical location (see S1 Fig for settlement locations plotted on a map of Uganda) | North-west Uganda, Adjumani District, close to the South Sudan border | Western Uganda, Kyegegwa District, between Fort Portal and Mubende | North-west Uganda, Yumbe District (West Nile), a few kilometres south of the South Sudan border |

local languages used by refugee communities in the settlements, and were able to provide informed consent to participate and had sufficient cognitive ability to be interviewed. Participants were excluded if the clinical team involved in the recruitment process judged that they were unable to take part in the study due to serious health conditions. Study information materials and consent forms were translated into languages commonly used across the settlements, including French, Swahili, Juba Arabic, Ma'di, and Lugbara.

## Data collection

Topic guides focused on the experience of living with advanced disease and continuity of care (e.g., those involved in delivering palliative care and supporting with symptom and disease management). The research team developed the topic guide in collaboration with patient and public involvement organisations through a workshop led by the African Palliative Care Association. Attendees included members of the Uganda Cancer Society, the Uganda Cancer Institute, the African Palliative Care Association Patient Involvement and Engagement Advisory Group, and members of the NCD Alliance. Feedback from the meeting included participants insisting on the need to know more about the lived experiences and challenges of refugee populations, rephrasing of questions to ensure readability for all education levels, and the need for all study materials to be translated and available in commonly spoken languages across the three sites, as outlined above. The topic guide was pilot tested and refined with three individuals living with advanced disease in the Bidibidi Refugee Settlement. These individuals were identified by community health workers supporting palliative care outreach, based on their ability to participate comfortably in an interview and their willingness to provide feedback on the clarity and appropriateness of the questions. Feedback from these pilot interviews was used to refine the wording and sequencing of questions. Pilot interview data were not included in the main analysis.

Face-to-face semi-structured interviews were conducted across all settlements. Data collection took place between July and November 2022 in Bidibidi Refugee Settlement, during April 2024 in Kyaka II Refugee Settlement, and between August and November 2024 in Nyumanzi Refugee Settlement. This phased approach reflected the time required to obtain site-specific ethical approvals and negotiate access agreements with implementing partners across each settlement. To ensure consistency across data collection phases, the same interview guides were used across all sites, interviewers were trained with a shared protocol, and reflexive field notes were maintained. During the data collection period, there were no substantive changes to national refugee health policy, settlement health service organisation, or palliative care provision models, as confirmed through regular consultation with district health leads and partner NGOs.

For all settlements, two researchers were involved in interviews, with one leading questions and the other taking notes. Participants could opt for a male or female interviewer. To promote consistency in data collection across sites and among interviewers with varying professional backgrounds, all interviewers participated in a joint training programme prior to commencing fieldwork. This included orientation to the study aims, standardised training in the use of the semi-structured interview guide (with practice interviews and peer feedback), and guidance on trauma-informed and culturally responsive interviewing. Interviewers used a shared, reflexive field-note template to document the contextual and relational features of each interview, and teams held regular debriefing meetings during data collection periods to support the ongoing calibration of their interviewing style and interpretive approach. A male clinical officer and a female psychologist conducted interviews at the Bidibidi Refugee Settlement. A male social anthropologist and a female social worker conducted interviews in Kyaka II Refugee Settlement. In Nyumanzi Refugee Settlement, interviews were conducted by a team comprising two female nurses, one male nurse, and a female social worker. Interviews were conducted with caregivers present in some instances, who provided additional details related to a participant's experience. All interviews were audio-recorded, transcribed verbatim and deidentified, and, where necessary, translated into English when local languages were used during the entire interview or through code-switching. Study materials (consent forms and interview guides) were translated into French, Swahili, Juba Arabic, Ma'di, and Lugbara to support shared comprehension across the settlements. Interviews were conducted in the language preferred by each participant, including English, Juba Arabic, Swahili,

Luganda, Kakwa, Bari, Lugbara, Keliko, and Dinka. Interviewers were fluent in the predominant local languages and culturally familiar, with the aim of fostering rapport and reducing linguistic hierarchy during interviews. A layered transcription and translation process was used: interviewers first transcribed and translated interviews into English, drawing on their linguistic and contextual knowledge. Translated transcripts were then reviewed by a second bilingual team member to assess semantic fidelity (ensuring that metaphors, idioms, and experiential descriptions were conveyed accurately) and cultural nuance (capturing tone, relational dynamics, emotional intensity, and references that may not translate directly). Where alternative interpretations were possible, these were discussed collaboratively to reach a shared understanding rooted in participants' intended meaning rather than literal substitution. Transcripts were not returned to study participants for comment due to limited time and resources.

### Data analysis

Thematic analysis was employed to analyse data, as this approach is suitable for exploring participants' lived experiences, views and perspectives [22]. Six steps of thematic analysis [22] were undertaken, involving familiarising the data, generating initial codes, searching for themes, reviewing themes, defining and naming themes, and producing the report. Data from the three settlements were coded using NVivo (R V.14.24.1). To enhance the reliability and validity of the data analysis, analyst triangulation was employed. This process involved two authors independently coding six transcripts, followed by a comparison that fostered consensus-building and enhanced the study's credibility. Because interviews were conducted in multiple languages, coding was undertaken using transcripts translated into English. Interviewers contributed to interpretive discussions during coding and theme development to clarify meaning across languages and reduce the influence of any single linguistic or professional standpoint. This collaborative cross-language review ensured that translation decisions reflected negotiated meaning rather than individual interpretation. Furthermore, regular cross-site reflexive discussions between interviewers and analysts supported interpretive consistency and helped mitigate the influence of interviewer backgrounds and the phased nature of data collection. To ensure data adequacy in this study, we followed the concepts of information power, conceptual depth, and theoretical sufficiency [23]. These concepts highlight key considerations for assessing data adequacy in qualitative research and provide a flexible and interpretive framework for determining when a dataset is sufficiently rich and relevant to address the research aims. We refined the study aim to ensure a clear focus and developed interview questions closely aligned with participants' lived experiences, enabling the collection of rich, relevant data, contributing to strong information power. The data also demonstrated conceptual depth, with participants offering detailed accounts of their physical and psychological experiences of advanced illness, as well as the challenges they faced while living in the settlements. Furthermore, the dataset showed both variation and consistency across different settlements, supporting the development of robust themes and indicating theoretical sufficiency in addressing the research questions meaningfully.

## Results

### Demographic data

A total of 44 participants were recruited, aged 19–80 years, with a mean age of 48 years. The majority of participants were female (n = 27; 56.25%). All participants were living with advanced conditions, including a range of non-communicable and progressive conditions. Table 2 presents the demographic and clinical characteristics of the sample, disaggregated by settlement.

The findings presented below draw on data collected across the three study sites—Nyumanzi, Kyaka II, and Bidibidi—and indicate where site-specific contributions shaped the analysis. The findings are organised thematically, reflecting patterns identified across the dataset and structured according to overarching themes and sub-themes.

**Theme 1: Displacement, illness progression, and the legacy of trauma.** This theme explores how participants' experiences of forced displacement, the development of illness over time, and exposure to traumatic events shaped their

**Table 2. Demographic characteristics of participants across three refugee settlements in Uganda.**

| Category | | Settlement | | | Overall study population |
|---|---|---|---|---|---|
| | | Bidibidi | Nyu-manzi | Kyaka II | |
| **Sex** | Male | 3 | 6 | 8 | 17 (38.6%) |
| | Female | 11 | 9 | 7 | 27 (61.4%) |
| **Diagnosis** | Breast cancer | 2 | 2 | 2 | 6 (13.6%) |
| | Heart disease | 2 | 0 | 4 | 6 (13.6%) |
| | Blood cancer | 0 | 3 | 1 | 4 (9.1%) |
| | Cervical cancer | 1 | 1 | 2 | 4 (9.1%) |
| | Liver disease | 1 | 2 | 1 | 4 (9.1%) |
| | Heart failure | 0 | 1 | 2 | 3 (6.8%) |
| | Kaposi's sarcoma | 0 | 1 | 1 | 2 (4.5%) |
| | Bone cancer | 2 | 0 | 0 | 2 (4.5%) |
| | Head and neck cancer | 2 | 0 | 0 | 2 (4.5%) |
| | Hypertension | 0 | 2 | 0 | 2 (4.5%) |
| | Cancer (unknown type) | 1 | 0 | 0 | 1 (2.1%) |
| | Stroke | 0 | 0 | 1 | 1 (2.1%) |
| | Hodgkin Lymphoma | 1 | 0 | 0 | 1 (2.3%) |
| | Penile cancer | 1 | 0 | 0 | 1 (2.3%) |
| | Eye cancer | 1 | 0 | 0 | 1 (2.3%) |
| | Hepatitis B and C | 0 | 1 | 0 | 1 (2.3%) |
| | Spondylitis | 0 | 1 | 0 | 1 (2.3%) |
| | Throat cancer | 0 | 0 | 1 | 1 (2.3%) |
| | Unknown disease | 0 | 1 | 0 | 1 (2.3%) |
| **Performance status (ECOG)** | Fully active, able to carry on all pre-disease performance without restriction | 0 | 0 | 0 | 0 (0.0%) |
| | Restricted in physically strenuous activity but ambulatory and able to carry out work of a light or sedentary nature | 0 | 3 | 8 | 11 (25.0%) |
| | Ambulatory and capable of all self-care but unable to carry out any work activities | 0 | 8 | 2 | 10 (22.7%) |
| | Capable of only limited self-care | 15 | 2 | 4 | 21 (47.7%) |
| | Completely disabled; cannot carry on any self-care | 0 | 1 | 1 | 2 (4.5%) |
| **Education** | None | 8 | 3 | 1 | 12 (27.3%) |
| | Primary | 5 | 8 | 7 | 20 (45.5%) |
| | Secondary | 1 | 3 | 5 | 9 (20.5%) |
| | Tertiary | 0 | 1 | 2 | 3 (6.8%) |
| **Marital Status** | Married | 5 | 10 | 9 | 24 (54.5%) |
| | Single | 1 | 2 | 3 | 6 (13.6%) |
| | Divorced or separated | 3 | 0 | 2 | 5 (11.4%) |
| | Widowed | 5 | 3 | 1 | 9 (20.5%) |
| **Caregiver present during the interview** | Yes | 12 | 15 | 1 | 28 (63.6%) |
| | No | 2 | 0 | 14 | 16 (36.4%) |

health and daily lives within refugee settlements. Narratives highlighted how prolonged uncertainty, fragmented care, and unresolved trauma contributed to a worsening of participants' physical and emotional wellbeing.

## (a) Forced displacement and ongoing insecurity

Participants from Nyumanzi and Kyaka II described how war, ethnic conflict, and targeted violence forced them to flee their home countries. These events were abrupt, terrifying, and left lasting psychological sequelae. One participant recalled, "*I came to live in Nyumanzi in settlement when the insurgency in South Sudan started.*" (*Kyaka-03, F, 35y, breast cancer*). Another stated, "*I ran away from Congo due to wars… I found myself here in Uganda.*" (*Kyaka-02, M, 39y, blood cancer*)

Accounts from Nyumanzi included vivid descriptions of repeated attacks, reflecting how persistent insecurity shaped the decision to seek refuge:

> "*That sometimes they [the Murule clan] can attack at night, and you have to run to the bush, and sometimes [you] must run because even after three days, they will come and attack [you] again at night, and you will have to run again [while they] shoot people and raid children. So they came to Uganda because of that, so because of the raid from Murule tribe of South Sudan, who could raid children, cattle, etc.*" (*Nyumanzi-11, F, 37y, blood cancer*)

## (b) Illness progression and fragmented care pathways

For some, illness began or worsened after arrival in Uganda. Participants described a slow progression of symptoms—often starting with pain, swelling, or bleeding—and repeated visits to health facilities before receiving a diagnosis:

> "*It was back in 2013 when I noticed that my left upper part of the abdomen had begun swelling and was painful, that prompted me to go to a clinic and get treatment.*" (*Bidibidi-11, M, 45y, liver cancer*)

Access to timely and consistent care was often interrupted by referral delays, unclear communication, or contradictory medical advice:

> "*It was now in 2017, after we had come to the camp, that I decided to go to Arua to be tested again, as the swelling was still there. In Arua, I was tested for many tests, which included Hepatitis B, which turned out to be positive; at this time still my stomach (abdomen) was still of normal size. I wasn't given any treatment still; rather, I came back to the camp and continued with my normal life… That was in 2020, around May, that's when I went to Swinga HC3 and explained myself; my stomach had begun swelling by then; I was rechecked for Hep B and was told it was the one causing me problems. Then, I was referred to Yumbe HC4, which in turn referred me to Arua (RRH) for surgery and further management. In Arua, the doctors said that my condition did not really necessitate surgery, and so, I was sent back to Yumbe HC4. It was around that same time that I started taking the medicines that I'm taking now.*" (*Bidibidi-11, M, 45y, liver cancer*)

Many participants only received a diagnosis at an advanced stage of illness. Several described living with uncertainty about their condition, which affected their ability to seek or begin treatment.

## (c) Trauma as antecedent and intensifier of illness

Some participants linked their current illness to earlier experiences of severe trauma, including witnessing killings, sexual violence, and the death of loved ones. These accounts reflect a perceived contributory relationship between psychological suffering and the emergence or worsening of physical illness:

*"I was first diagnosed with epilepsy. This disease was because I witnessed my parents being cut to death with pangas. It started developing into stress… my baby also died mysteriously. So, whenever I sit alone, I remember all those incidents. I was taken to hospital after rape, I was told by the doctors that I had issues with my brain, and it is what causes epilepsy. I asked the doctors what the cause of this disease could be, I was told that I can get it without any cause, and I started receiving tablets. I have taken tablets for long in an attempt to recover, but sickness keeps on worsening. And this is how I got cancer, and my leg became lame. I was raped in Congo to the extent of my uterus coming out. After my uterus coming out, it was infected with dust and dirt. I took long sharing it out because I was mentally disturbed."* (Kyaka-13, F, 30y, cervical cancer)

Past traumas continued to affect participants after resettlement, with some reporting intrusive memories, emotional distress, and depression:

*"When I reached here in Uganda, I still experienced the first challenges of overthinking about the past."* (Kyaka-15, M, 39y, heart disease)

*"There is something still traumatising my mind, it's something shameful, but let me say it out… the man came after raping my wife and forcefully told me to lick his penis [he started shedding tears…] it was so hurting… it hurt me seriously."* (Kyaka-12, M, 40y, heart disease)

These experiences illustrate the amassing of violence, displacement, and healthcare delays. Trauma was not only a source of historical suffering but a continuing presence in participants' lives, reinforcing distress and shaping how they experienced and responded to illness.

**Theme 2: Impact of illness while living in settlements.** This theme explores the multidimensional effects of advanced illness on participants' lives across the three settlements, including physical limitations, economic hardship, emotional distress, stigma, and strained interpersonal relationships. Illness was described not only as a clinical condition but as a deeply disruptive force that reshaped everyday life, strained family dynamics, and diminished participants' sense of self and social value.

**(a) Physical limitations and loss of function**

Across all three sites, participants reported progressive physical deterioration that restricted their ability to perform daily tasks such as walking, fetching water, cooking, or working. These impairments significantly altered their roles within families and communities:

*"I used to fetch my water, I used to even dig and fetch firewood for myself, cooking, all this I can't do now because of this sickness, even travelling long distances I stopped because of the illness."* (Bidibidi-08, F, 66y, head and neck cancer)

Participants also expressed feelings of being reduced to a state of physical passivity, which contributed to a loss of meaning and erosion of self-worth. One described this as: *"Now the sickness has reduced me to nothing. I feel like I am just here, I am useless."* (Nyumanzi-07, M, 66y, heart failure).

Where mobility impairments existed, participants identified assistive devices such as wheelchairs as urgently needed but unavailable. In addition to mobility loss, participants described a range of symptoms—including shortness of breath, chest pain, dizziness, and headaches—that compounded their physical vulnerability and increased dependence on others.

*"The major support she was expecting herself was to get a wheelchair so that she can be moving out, like when they are going to bath her. Also when it is dry like this, she can also get out in that wheelchair so that she can get fresh air."* (Nyumanzi-01, F, 70y, hypertension)

**(b)   Economic insecurity and disrupted livelihoods**

The inability to work due to illness was one of the most reported consequences, with many participants linking their deteriorating health to deepening poverty and food insecurity:

*"Because I am always bedridden, not working, getting money has become hard for me." (Nyumanzi-13, F, 75y, cervical cancer)*

*"I'm ever here. I don't work. It's only my husband and children who go to search for jobs and get us what to eat." (Kyaka-04, F, 34y, cervical cancer)*

Participants frequently spoke of their loss of provider roles, which they saw as integral to their dignity and to the survival of their families.

**(c)   Emotional distress and psychological burden**

In addition to material deprivation, participants articulated a deeper psychological and existential need: a need to be seen, recognised, and valued as human beings despite the suffering they were experiencing. This need was closely tied to feelings of lost identity and diminished social roles. As one participant described: *"Before I was someone who could help others. Now, I am the one who is helped, I have nothing. Even my children look at me with pity." (Nyumanzi-08, F, 49y, Spondylitis)*.

Illness triggered persistent psychological distress, described through feelings of depression, fear, worthlessness, and helplessness. These emotional responses were exacerbated by participants' inability to meet familial and community expectations:

*"It has impacted on my mental health; it has made me to think too much because I have children, and I have told you my children are too young, so it has made me depressed all the time; if I did not have this illness, I would have gone to garden to go and look for something for my children to eat, but because of this sickness I am ever down, and it has made me mentally ill." (Nyumanzi-04, M, 44y, liver disease)*

Some spoke about a profound desire for death or passive acceptance of death as inevitable, expressing the depth of their psychological and existential suffering. One participant stated: *"I feel like I am just staying here waiting for death." (Bibi Bidi-03, F, 70y, breast cancer)*.

**(d)   Stigma, isolation, and social rejection**

Illness was also a source of stigma and exclusion. Participants described being insulted, ignored by relatives, or avoided by community members due to the visible signs of their condition:

*"I know of relatives who come from Juba, stay around but fail to reach me here, that really troubles my heart, it makes me feel really useless and worthless to them." (Bidibidi-06, F, 63y, cervical cancer)*

*"Wherever I go, people keep on insulting me because of epilepsy. It really hurts me if I think about it. (She became emotional…)" (Kyaka-13, F, 30y, cervical cancer)*

*"I swear, living with this swelling has changed my life. I'm limited to staying mainly at home, avoiding public places given that whenever I move around, people start backbiting me given the swelling that I have on my face… this makes me overthink a lot about my sickness, my heart pains from this, giving me a lot of stress and depression from the overthinking." (Bidibidi-01, F, 35y, bone cancer)*

## (e) Family strain and role disruption

Illness often altered participants' relationships with spouses, children, and extended family, sometimes resulting in abandonment or marital conflict. Participants outlined illness as the cause of a spouse leaving them, alongside estrangement and hostility from family. Some participants felt they had become a burden, leading to tension and blame within the household:

> "My family is very stressed. The children cry that I am going to die and leave them here, so does my husband. He even sent me away. He doesn't want to see me in this state. He says that whenever he looks at me, he sees a dead person. In fact, he sent me to the camp—that if I'm to die, better I die from here and maybe in case of other things, people would help from here." (Bidibidi-06, F, 63y, cervical cancer)

> "I always receive a lot of complaints from my mother concerning my sickness, which doesn't heal. She complains the way she has spent a lot of money on my sickness the way she is tired of me. I develop a lot of thoughts… she is tired of me asking money for buying tablets." (Kyaka-06, M, 30y, heart failure)

Participants also reported losing contact with friends and relatives:

> "Every friend of mine ran away from me ever since I got this sickness." (Kyaka-05, M, 33y, breast cancer)

## (f) Loss of caregiving role and provider identity

For many, one of the most painful aspects of illness was the loss of their ability to provide for and protect their families. This was particularly acute for parents with young children:

> "My worry is my children are still very young, and there is no one to support them in their education, and there is no one to support my wife and now, if I'm down, who is going to support them in the future, who is going to raise them up." (Nyumanzi-04, M, 44y, liver disease)

> "My responsibilities have been shifted to the young girls (pointing at the girls in the compound); things like fetching water, collecting firewood is supposed to be my responsibility. However, now that I'm unable to, this workload ends up shifting down to the young ones." (Bidibidi-01, F, 35y, bone cancer)

> "I used to go and search for them what to eat… Some days, I used to go for digging and get food… But nowadays I don't go for such activities. My children were studying by then. I cannot afford their school fees currently because I don't work." (Kyaka-09, F, 53y, breast cancer)

Losing their roles as caregivers or providers was a profound source of distress, reflecting a perceived erosion of identity and self-worth intimately connected to their sense of dignity.

**Theme 3: Managing illness in settlements.** This theme outlines the various strategies participants employed to manage their illness within the constraints of life in the settlement. These included support from palliative care teams, informal networks of family and religious groups, personal coping strategies, and adaptations to resource limitations. Participants also described the limitations of support and their perceptions of palliative care, highlighting both its value and its constraints.

## (a) Informal and community support

Participants commonly relied on support from relatives, neighbours, friends, and religious communities to meet daily needs or cope emotionally. This support was often minimal, described as help with necessities, encouragement, or prayer:

*"They [participant's children] buy me necessities that I use at home, for example, shoes, sugar and sometimes charcoal to cook." (Bidibidi-06, F, 63y, cervical cancer)*

*"It's only food because she [participant's friend] is also struggling to survive. She helps me with food, and maybe her children help to bathe me… there is not any other support." (Kyaka-01, F, 67y, throat cancer)*

*"They [religious leaders] give advice, they say many people are suffering like you, sometimes it's temptation—one day God will relieve you if you are overthinking, and this children will not be okay." (Nyumanzi-05, F, 52y, heart failure)*

Several participants also highlighted the care they received from family members, particularly daughters and female relatives, who helped with hygiene, mobility, and emotional support:

*"She is the one with her [the participant's] daughters, who are bathing her, carrying her out, washing her clothes because she doesn't move out. She can just urinate inside… washing her clothes—she is the one who is taking care of her, they are the one supporting with her daughters, the two daughters who are outside." (carer; Nyumanzi-01, F, 70y, hypertension)*

*"I always receive counselling whenever I come here in Bujubuli. I'm told not to overthink due to the situation. It's general situation, so we shouldn't overthink. Problems come and go. We also get counselling in church, which encourages us to move on with life regardless of the situation." (Kyaka-07, F, 44y, heart failure)*

Some participants described relying on herbal remedies or traditional practices when formal medical care was unavailable. One participant noted: *"I was using herbs because the hospital kept sending me back without medicine." (Nyumanzi-08, F, 49y, spondylitis).*

## (b)  Perceptions of palliative care

Many participants—particularly those in Kyaka and Nyumanzi—valued the support received from the palliative care team, citing the provision of medicine, psychosocial support, and reassurance:

*"I don't have any blame for the organisation that provide us with palliative care. We always get the available medicine, and they recommend buying the one they don't have." (Kyaka-03, F, 35y, breast cancer)*

*"She is really happy with the palliative care team because at least today she is going to sleep well because they have come and talked to her, and they have really consoled her. And she [is] seeking God to protect the palliative care team in the work they are doing in the community, and if possible, God should really bless them such that they can take care of the children that she is having at home." (carer; Nyumanzi-10, F, 48y, liver disease)*

However, some participants reported a limited understanding of what palliative care involved, viewing it primarily through the lens of medicine distribution rather than holistic symptom management and psychosocial care:

*"Will not be able to say the difference because she [the participant] use to mix them so she cannot be able to differentiate." (carer; Nyumanzi-12, F, 80y, breast cancer)*

*"She wants to know what exactly the palliative care team does so that in case she has other challenges, she can make the request." (carer; Nyumanzi-09, F, 19y, unknown disease)*

Others perceived the main distinction between hospital-based care and palliative care as financial:

*"In the hospital, you buy drugs, but the one from the palliative care team is free." (Nyumanzi-13, F, 75y, cervical cancer)*

Several participants believed that the scope of support provided by palliative care teams was limited, primarily addressing pain relief and offering verbal comfort, but not fully meeting other practical or medical needs:

*"But due to lack of their capacity [the palliative care team] to support other needs of mama [the participant]…" (Nyumanzi-01, F, 70y, hypertension)*

*"They [the palliative care team] only gave her psychosocial support and oral liquid morphine and other medical support is what she has been receiving from the palliative care team." (Nyumanzi-07, F, 36y, breast cancer)*

### (c) Individual coping strategies

Despite the challenges, participants described ways they coped with pain and emotional distress. Some turned to prayer and spiritual practices, with others using social interaction as a way of alleviating suffering:

*"When my pain comes, I normally go to my neighbours, and we converse. There's something about the conversation that calms me down and takes my mind off the pain. Eventually, I feel better." (Bidibidi-01, F, 35y, bone cancer)*

Yet for many, existential suffering remained unaddressed. Several participants reported that despite receiving support from palliative care teams or community networks, a sense of hopelessness endured. One participant shared: *"They come, they talk to me, I thank them, but inside I am still not okay. I just wait here. My body is tired." (Nyumanzi-06, F, 60y, breast cancer)*.

Across accounts, participants' perceptions of worsening health were closely linked to the fragmented and intermittent nature of care, including irregular review appointments, varying availability of medicines, and reliance on multiple providers across camp and referral facilities. This discontinuity contributed to a sense that deterioration was both expected and unavoidable.

**Theme 4: Challenges and needs in settlements.** This theme outlines the intersecting material, systemic, and structural challenges participants faced while living with serious illness in refugee settlements. Across all three sites, participants described critical shortages of basic needs, limited access to healthcare, financial strain, and systemic barriers, including language barriers, transportation issues, and disrupted continuity of care. These challenges shaped both how participants experienced illness and their approach to seeking care.

### (a) Unmet basic needs

The most commonly reported challenge across the settlements was the widespread lack of food, shelter, clothing, and other essential resources, including education for the children and young people in the families of participants. Participants described daily struggles to meet even their most basic survival needs:

*"One can say the food ration is there in the camp; the challenge is that it's not enough to support our feeding needs, in fact the food given doesn't even last a month but rather ends midway; that's what really is challenging." (Bidibidi-11, M, 45y, liver cancer)*

In some cases, illness prevented participants from maintaining and rebuilding of damaged shelters, increasing their dependency on others:

*"Another challenge is hunger and poor shelter. The house I used to sleep in got destroyed by weather. The well-wishers provided this one where I'm staying currently because I had no energy to construct a new house of my own." (Kyaka-10, M, 45y, Kaposi's sarcoma)*

Participants consistently emphasised the absence of meaningful external support, particularly financial support, from NGOs, community organisations, or formal assistance systems:

*"There is not any support from the church; there is not any support from organisations. So, she is herself [the wife of the son] who is supporting her with the relatives who are at home that are supporting her." (Nyumanzi-01, F, 70y, hypertension)*

## (b) Health system barriers and gaps in care

Participants across all three settlements reported significant difficulties accessing healthcare. These included inconsistent follow-up, long delays, and unclear communication. Several accounts described bureaucratic delays and inefficiencies:

*"A classic example is the first time that I went to Mulago; due to some errors in my documentation, I wasn't attended to but rather sent back to Kuru to rectify my paperwork and that delayed the process of getting support." (Bidibidi-10, F, 30y, head and neck cancer)*

*"I always come here [health centre in the settlement], but the doctor who is on duty keeps telling us if we don't have any appointment with the doctor, we should go home and wait for the call in order to go to Mulago. That's what I'm still waiting for ever since January." (Kyaka-07, F, 44y, heart failure)*

Participants also described difficulties arising from staff turnover and inconsistent care:

*"The situation we go through to meet the caretakers in the hostels before distributing us to different doctors is really challenging. This is because they keep changing doctors. You communicate with this doctor today, the following day you find a different one, which makes it hard for them to follow up with treatment we receive." (Kyaka-15, M, 39y, heart disease)*

Language differences between healthcare staff and patients created major obstacles, particularly for Congolese refugees who spoke Swahili rather than English or Luganda:

*"The only challenge we get is language barrier. Doctors in Mulago don't know Swahili. They expect us to speak English or Luganda which we don't know and that's how we miss treatment. We don't get interpreters from the hostel where we stay… we fail to explain the detailed information concerning our diseases due to lack of an interpreter. If by chance you find the doctor you can communicate with, you will get tablets and [they will] recommend buying other tablets missing or you get them from Bujubuli." (Kyaka-07, F, 44y, heart failure)*

Long travel distances to reach urban hospitals and referral centres were also reported as a major barrier to accessing care, particularly for those with physical impairments or limited funds:

*"We are put in one hostel in Kanyanya and in the morning we are taken to different hospitals. I'm always taken to Kiruddu. It can take like three or two hours while in the vehicle before reaching Kiruddu… They always remain with our telephone numbers in order to call us when it's time for going for review. When I'm called, I leave my home at around 7:00pm to go and sleep over at the health Centre. We set off to Mulago at 7:00am in the morning." (Kyaka-14, M, 65y, heart disease)*

## (c) Inability to afford medicines or follow treatment advice

Where medicine was prescribed, participants were often unable to purchase it due to a lack of financial means. These economic constraints led to untreated symptoms, self-neglect, and trade-offs.

*"The support I get from hospital is only when I go, they will test, and after testing they say, we do not have this medicine; you go to the clinic and go and buy and if I do not have money I just come and sit." (Nyumanzi-04, M, 44y, liver disease)*

Resource limitations also meant that, often, additional suggestions from health professionals, such as advice regarding diet and care, could not be adhered to:

*"Doctors recommended not to eat foods containing fats, salt among others. The doctors told me that my veins are covered with fats, so I have to reduce eating foods with much fats, I should not eat meat. So, I have to eat greens. The doctor recommended fruits like watermelon, fish, which are costly. What will I buy fish with, yet I don't work? That's a challenge as well… I need help with food." (Kyaka-12, M, 40y, heart disease)*

Participants frequently linked their declining physical condition and increasing emotional distress to these systemic gaps in continuity of care, alongside limited follow-up after referral and inconsistent access to essential medicines. This contributed to a widespread perception that health deterioration was inevitable and that opportunities for stabilisation or symptom control were limited.

To support the depiction of cumulative vulnerability across psychosocial, economic, physical health, and structural domains, Fig 1 presents a conceptual model derived from the study data.

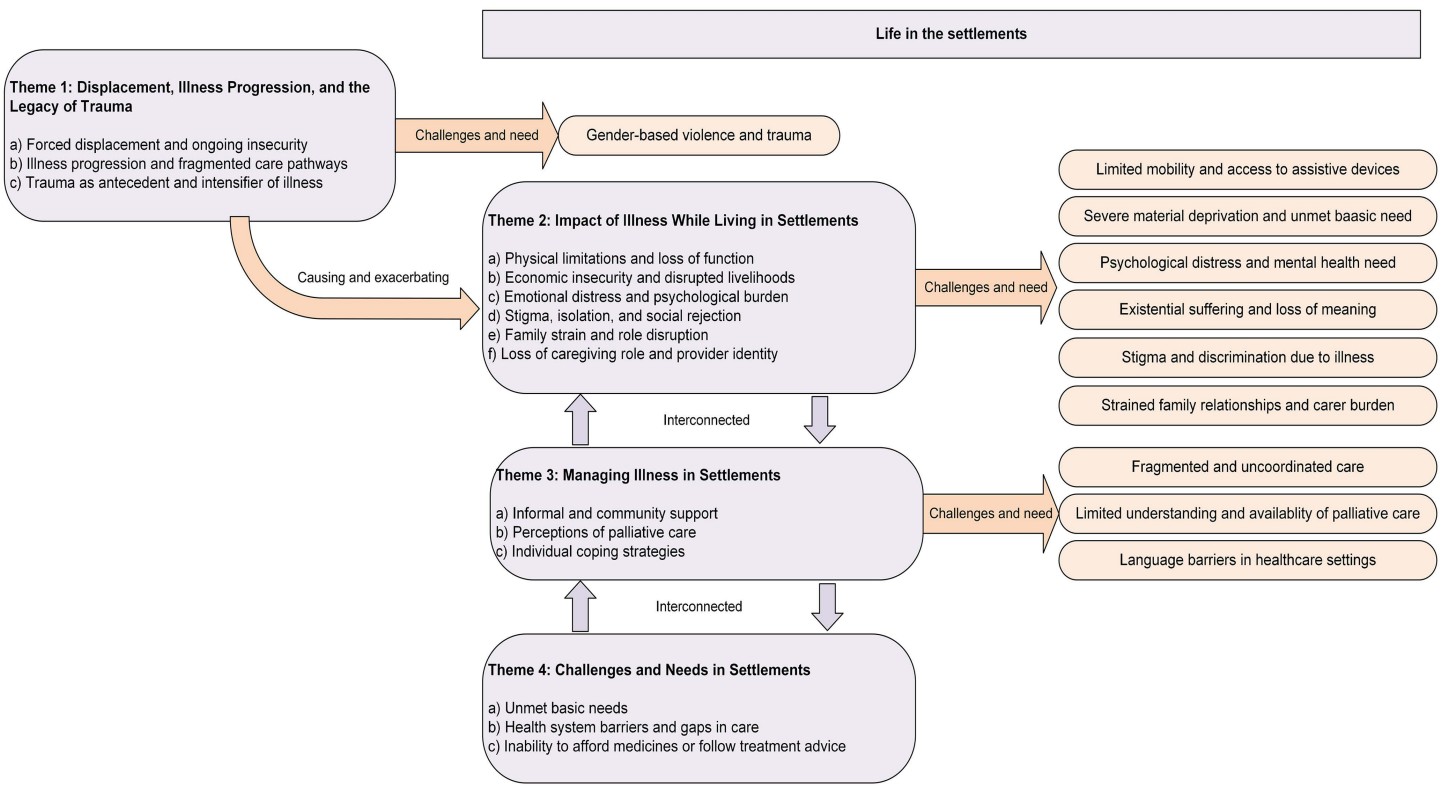

**Fig 1. Conceptual model synthesising themes identified in this study.**

## Discussion

Our findings highlight the multifaceted experiences of people living with advanced illness in three refugee settlements in Uganda. Although palliative care aims to address the complex and changing physical, emotional, social, and spiritual needs of people with life-limiting conditions [24], participants across all three sites in this study perceived the current care as insufficient in meeting even basic needs, thereby exacerbating their daily suffering. Their accounts reflected multiple, intersecting challenges as echoed in the existing literature: limited access to healthcare, insufficient family and institutional support, stigma, financial hardship, and linguistic and cultural barriers [25]. These challenges compounded the impacts of serious illness, contributed to a sense that deterioration was inevitable, underscoring the neglect of displaced populations in palliative care provision [26]. Alongside these shared structural challenges, the study also illuminated more granular, context-specific factors shaping the delivery of care, including fragmented provider interaction, discontinuity of care across providers, gender-based violence, and sustained psychological distress.

Participants' accounts demonstrated a close relationship between their perceived deterioration and the fragmented nature of care across providers within and beyond the settlement. Difficulties obtaining consistent review, limited availability of essential medicines, and unclear or delayed follow-up contributed directly to participants' sense that deterioration was inevitable. In this way, worsening physical and emotional health was experienced not only as clinical progression, but as something shaped by the organisation and continuity of care. The Humanitarian–Development–Peace (HDP/3D) Nexus offers a lens for understanding these patterns, emphasising the alignment of immediate humanitarian relief with longer-term health system strengthening and governance in protracted crises [27,28]. This is particularly relevant for palliative care, where needs are not always short-term. For example, while some individuals with rapidly progressive cancers may require intensive support over weeks or months, others—such as those living with advanced HIV, heart failure, or diabetes-related organ failure—may experience fluctuating periods of stability and decline that require sustained, adaptive forms of support [29,30]. In many settlement health systems, however, service delivery remains structured around emergency-oriented triage established during the acute phase of displacement, where prioritisation is based on immediacy and threat to life [31]. As displacement becomes protracted, the continued application of these emergency models results in systematic under-recognition of chronic and advanced illness needs, particularly given the rising burden of non-communicable disease and multi-morbidity among displaced populations [32,33]. This represents a distinct form of institutional exclusion, separate from broader structural violence, that emerges when systems built for rapid crisis response are not reconfigured for long-term care or with health system resilience in mind [34,35]. When considered alongside Camp Management frameworks, which outline how responsibilities for coordination and service delivery are distributed across humanitarian, governmental, and community actors [36], the fragmentation participants reported may be structurally produced rather than incidental. Strengthening palliative care in settlements, therefore, requires coordinated approaches that operate across systems and over time, recognising that many people with advanced illness will require support that is sustained, relational, and embedded within routine health system functioning. This aligns with evidence from palliative care programmes in Uganda and neighbouring countries, where longitudinal symptom management, psychosocial support, and home- and community-based care models are feasible and effective [29,30].

To provide a practical synthesis of these findings, we present two outputs from the data generated in this study. Fig 1, in the Findings section, presents a conceptual model derived from the study data. Table 3 builds directly from participants' accounts to map the key needs identified in this study and align them with potential actions and required resources by stakeholders, grounded in existing evidence and best practices. It provides practical action to inform responsive, context-sensitive palliative care approaches in humanitarian settings. This synthesis of findings and existing evidence offers a policy- and practice-relevant tool, bridging insights from this study with actionable interventions. Each challenge is paired with a proposed response suitable for humanitarian or low-resource settings, such as expanding decentralised drug distribution, integrating family caregivers into care models, or improving health worker training on trauma-informed approaches [37]. Where relevant, these recommendations are grounded in established evidence, including WHO

**Table 3.** Proposed context-appropriate responses to challenges identified in this study, aligned with multisectoral coordination frameworks relevant to protracted refugee settings.

| Key Challenge | Potential response | Resources required | Best practice examples that may warrant exploration or further development in the context of refugee settlements |
|---|---|---|---|
| Fragmented and uncoordinated care | Local healthcare providers and community-level actors: Establish integrated referral pathways involving local health facilities, NGOs, and community workers, supported by routine communication and shared patient records to enhance continuity and coordination. | Referral protocols and tools; Training and workshops for staff and community health workers; Shared patient records; Communication systems; Transport support; Monitoring and evaluation mechanisms. | Care coordination can reduce fragmentation and improve outcomes for fragile, conflict-affected and vulnerable communities [38]. |
| Limited understanding and availability of palliative care | Community-level actors: Deliver community-led education programs through trained community health workers to raise awareness of palliative care and integrate basic palliative services into existing primary healthcare structures. | Training materials and workshops for community health workers; Supportive supervision and mentorship; Transportation and communication tools; Educational materials in local languages; Monitoring and evaluation systems; Funding for outreach activities and stipends. | Integrated community palliative care models can improve equity and access [39]. The use of digital systems for determining palliative care provision continues to evolve and provide opportunities for monitoring and evaluation [40]. An example is the Health Management Information System (HMIS) developed by the Ministry of Health of Uganda, which collects data on people receiving palliative care interventions, providing insights for policy formulation, decision-making, and palliative care research. |
| Severe material deprivation and unmet basic needs | Local authorities, NGOs and community-level actors: Implement cross-sectoral partnerships involving humanitarian agencies, local government, and NGOs to provide essential food, shelter, and basic commodities, prioritising vulnerable groups within refugee settlements. | Cross-sector coordination mechanisms; Mapping and data systems to identify and prioritise vulnerable groups; Logistics support for distribution; Funding for basic needs; Monitoring and accountability systems to ensure equitable distribution. | The Sphere Handbook [36] outlines minimum standards for providing basic needs in humanitarian settings. |
| Psychological distress and mental health needs | Local healthcare providers: Train general healthcare workers and non-specialist providers in culturally sensitive mental health approaches, establishing clear referral mechanisms to specialist support, and integrating routine mental health assessments into primary care visits. | Culturally adapted training materials and workshops; Referral guidelines and directory of local mental health services; Screening tools for routine mental health assessments; Supervision and mentorship structures to support non-specialists; Monitoring systems to track referrals and outcomes. | The Friendship Bench model supports task-sharing mental health care in LMICs [41]. |
| Existential suffering and loss of meaning | Religious leaders, local authority and NGOs: Integrate culturally grounded spiritual and narrative interventions into palliative care services within refugee settlements by working with faith leaders, elders, and trained community health volunteers. These interventions should be delivered alongside clinical care—either during routine home visits or at community health points—providing structured opportunities for patients to explore meaning, identity, and grief through storytelling, prayer, and reflective dialogue. By embedding these practices into existing palliative care frameworks, such as home-based care or symptom review clinics, they can help address spiritual distress and support psychosocial wellbeing, even where specialist providers are limited. | Training materials and guidance manuals for culturally adapted interventions for healthcare workers; Monitoring and supervision systems. | Dignity-based narrative interventions reduce existential suffering and enhance spiritual peace among patients with advanced cancer [42]. These approaches can be adapted for humanitarian settings and delivered in tandem with basic palliative care by trained lay providers or volunteers, particularly where specialist spiritual care is unavailable. |

*(Continued)*

**Table 3.** (Continued)

| Key Challenge | Potential response | Resources required | Best practice examples that may warrant exploration or further development in the context of refugee settlements |
|---|---|---|---|
| Strained family relationships and carer burden | Caregivers, healthcare workers, community-level actors, and NGOs: Structured training, financial, and material support for family caregivers could be explored, alongside the establishment of caregiver peer-support groups and respite care options within existing community health frameworks. | Structured caregiver training packages; Materials and job aids such as visual guides, home care tools, checklists, and low-literacy resources; Peer support group structures; Monitoring tools to assess caregiver burden, well-being, and service impact. | Caregiver training and support can help reduce the burden and distress [43]. |
| Stigma and discrimination due to illness | Community leaders and influencers, NGOs and local authorities: Conduct community dialogues and awareness campaigns led by respected local figures to challenge stigma, facilitate peer support groups, and build inclusive social environments for those with chronic illnesses. | Training for local facilitators on stigma, advance illness awareness, and inclusive language; Stipends or recognition for peer leaders and facilitators Monitoring tools to assess changes in community attitudes and social inclusion. | Community-based stigma interventions improve inclusion and mental health [44]. |
| Language barriers in healthcare settings | Multilingual healthcare workers, local authorities, community-level actors: Explore options for deploying multilingual health staff and community interpreters in refugee healthcare facilities, and develop pictorial or simplified multilingual information materials to enhance health literacy and improve patient-provider communication. | Recruitment or training of multilingual health staff and community interpreters; Guidelines and protocols for using interpreters in clinical settings (confidentiality, neutrality, accuracy); Translation services or partnerships for accurate and timely content production; Monitoring tools to evaluate communication effectiveness and patient understanding. | Visual tools enhance patient communication in linguistically diverse settings (e.g., [45]). |
| Limited mobility and access to assistive devices | Local authorities, NGOs and caregivers: Ensure the provision of assistive devices, such as wheelchairs, through local health facilities, supported by training caregivers in their use, maintenance, and integration into routine home-based care programs. | Appropriate assistive products; Training materials and sessions for caregivers. | The WHO Priority Assistive Products List supports the provision of mobility aids. |
| Gender-based violence (GBV) and trauma | Local mental health services and NGOs: Integrate GBV screening and trauma-informed care into routine health checks, enhance referral pathways to protection services, and provide specialised psychosocial support to survivors through local healthcare and NGO partnerships. | Training for health workers on GBV identification, confidentiality, and trauma-informed approaches; Standardised GBV screening tools integrated into routine care; Referral directories and protocols aligned with local protection services; Psychosocial support services staffed and resourced. | Integrated GBV screening and referral models in HIV and general healthcare settings have demonstrated feasibility and acceptability, with high detection rates and meaningful outcomes, such as receipt of counselling and ongoing support [46,47]. |

guidelines, systematic reviews, and published interventions from comparable contexts. This table is designed to inform multisectoral planning across health, protection, and social care sectors, providing a roadmap for more responsive and resilient palliative care in displacement settings.

Evidence suggests that fragmented care is a pervasive issue in end-of-life settings and is associated with increased mortality, higher complication rates, and higher costs [48]. In refugee contexts, systemic fragmentation is further reinforced by policy and legal exclusions, limited resources, and poor continuity of care. Participants described cycles of referral

delays, unmet follow-ups, and unclear treatment plans that contributed to significant uncertainty and distress. Research indicates that strong patient-provider relationships and integrated records are crucial in reducing such fragmentation and enhancing care outcomes [49]. In line with WHO and Sphere guidelines, there is a need for integrated referral systems, service mapping, and improved coordination between healthcare providers, NGOs, and community actors to ensure continuity and responsiveness.

Participants frequently described how the burdens of illness disrupted family structures and strained relationships, resulting in abandonment or emotional withdrawal. These relational ruptures were often intertwined with psychological distress and stigma. These findings align with earlier work [50] that emphasises the crucial importance of involving family caregivers in palliative care assessments and support frameworks. The wellbeing of caregivers and families is an essential component of sustainable palliative care, necessitating structured inclusion in care planning and support mechanisms from NGOs.

Experiences of gender-based violence—both as a driver of displacement and as a continuing feature of life in the settlements—emerged strongly in participant narratives. Survivors of sexual and psychological abuse linked their experiences to ongoing physical and mental health deterioration, consistent with the literature on GBV's impact on morbidity and psychosocial wellbeing [51–54]. Existing frameworks for addressing GBV in African humanitarian contexts primarily focus on legal protection and emergency services [55]; however, integration into palliative care remains minimal. Addressing this requires trauma-informed, survivor-centred approaches embedded within broader care delivery models.

Psychological distress was a near-universal experience among participants, expressed through persistent worry, low mood, fear, and, in some cases, suicidal ideation. These emotional burdens were shaped not only by the experience of advanced illness but also by the wider adversities of displacement, including the loss of home, separation from family, precarious livelihoods, and uncertainty about the future. This psychological distress was further compounded by the limited availability of essential palliative interventions in Uganda, such as access to opioids for pain management and specialist palliative care services. Those who are living in the refugee settlements already face structural barriers to basic health services, placing them at greater risk of unmanaged suffering and a profound loss of dignity. These findings are consistent with syndemic frameworks [7], which conceptualise the interaction of chronic illness and social adversity as mutually reinforcing, rather than independent phenomena. Models such as the five pillars of post-conflict recovery [56]—safety, social bonds, justice, identity, and meaning—offer a useful structure for conceptualising how displacement and illness destabilise psychosocial functioning. Although there is limited evidence on psychological interventions embedded within palliative care for displaced populations [57], recent trials of culturally adapted mental health interventions suggest that they are both feasible and effective in low-resource and humanitarian settings [58].

Many participants also described profound existential suffering. Several articulated a sense of merely "waiting for death"—a state of suspended life characterised by hopelessness, loss of dignity, and erosion of life's purpose. This existential dimension of distress, though frequently overlooked in humanitarian responses, is central to the ethos of palliative care to address total suffering [59,60]. Existing literature underscores the significance of meaning-making, identity continuity, and spiritual support for individuals with life-limiting illnesses, particularly those facing compounded forms of marginalisation [61]. Yet, in refugee settlements where clinical, spiritual, and psychosocial services are often fragmented or absent, access to such existential support is minimal. Developing palliative approaches that incorporate culturally relevant spiritual care, narrative therapies, and dignity-conserving interventions is crucial for responding to the full scope of needs in these settings, especially when curative care is unavailable or inaccessible. To build resilient and person-centred palliative care in refugee settings, psychological support must be integrated as a core service pillar. This requires not only training and supervision of healthcare workers, but also early-stage planning for sustainable mental health systems that can function effectively through protracted crises. As recommended by The Sphere Handbook [36], such efforts should be locally led, culturally grounded, and inclusive of both formal and informal support networks.

## Limitations

This study has several limitations. First, the findings reflect lived experiences within three specific Ugandan settlements and may not be transferable to all displacement contexts. Second, trauma was prominent, but we did not systematically screen for or record trauma in this study. It may have arisen where people were comfortable to share and disclose it, and some experiences may not have been reported. As such, we have not reported explicit types of trauma, but instead draw together and report on the data that was collected during interviews. Third, the study did not include healthcare providers, thereby limiting the triangulation of perspectives. Fourth, data collection occurred in phased periods across 2022 and 2024. While this enabled the inclusion of multiple settlements, it may also have introduced contextual variation over time. However, national refugee health policy, settlement health system organisation, and palliative care outreach models remained stable during this period, and shared interview guides, training protocols, and reflexive field notes were used to maintain consistency across phases. Fifth, the study did not compare the participants' experience, such as stigma, healthcare and violence, before and after living in the refugee settlements, which may have limited understanding of what challenges were newly introduced, intensified or already present prior to displacement. Sixth, variability in interviewers and data collection methods across sites may have introduced inconsistencies. While cross-language interviewing and translation may introduce some loss of nuance, this was mitigated through verification of translations, shared English-language coding, reflexive field notes, and cross-site analytic dialogue. However, we acknowledge that some fine-grained linguistic tone and affect may not be fully recoverable in translation.

## Implications

This study provides urgent evidence to inform the design and implementation of palliative care in displacement contexts. Findings highlight the need for coordinated, integrated care models that meet both medical and psychosocial needs. Whilst palliative care is recognised within global humanitarian health standards (e.g., the Sphere Handbook [36] and WHO guidance on palliative care integration in humanitarian emergencies and crises [62]), implementation remains limited and uneven. Realising these standards in practice requires reliable access to essential medicines (including opioids for pain relief), training and mentorship for health workers at multiple levels, integration of palliative care into routine primary care and referral pathways, and coordination across humanitarian and national health systems. These efforts should be grounded in contextually meaningful psychosocial, spiritual, and community-engaged approaches [62].

At the same time, NGOs and service providers must prioritise the inclusion of caregivers, trauma-informed care, and the integration of mental health and protection services. Table 3 provides a practical synthesis that maps the needs identified by participants to actionable responses and resource requirements. This offers a feasible route to strengthening palliative care in refugee settlements without establishing new or parallel systems by improving coordination between existing humanitarian and national health system structures. Future research should explore scalable models of culturally adapted psychological interventions within palliative care and rigorously evaluate their impact on displaced populations. Ultimately, effective palliative care in humanitarian settings must be relational, responsive, and embedded in multisectoral efforts to uphold dignity and reduce suffering in contexts of extreme adversity.

Participants chose to disclose experiences of sexual violence, despite the sensitivity of the topic. This suggests the importance of creating a safe, empathetic interview space where participants feel heard without judgment. The interviewers' empathetic, sensitive, professional, and ethical manner could play a key role in building trust, allowing participants to speak about deeply personal experiences [63]. This further underscores the value in adopting trauma-informed approaches in palliative care practice and research [37], particularly in humanitarian settings.

## Conclusions

This study offers critical insights into the lived experiences of displaced individuals with advanced illness in three refugee settlements in Uganda, revealing how structural deprivation, fragmented care systems, and cumulative psychosocial

trauma interact to shape profound vulnerabilities. Participants' narratives underscore the imperative to integrate palliative care into humanitarian responses—not as a peripheral service, but as a fundamental component of health system strengthening. By distilling key findings, this manuscript provides practical guidance on evidence-informed approaches to enhance palliative care delivery in refugee settings. Addressing the multidimensional challenges identified requires multi-sectoral strategies that combine medical care, psychosocial support, and social protection, delivered in culturally sensitive and contextually appropriate ways. By centring the voices of displaced people with advanced illness, this research contributes to the growing call for more equitable and person-centred models of care that respond to the syndemic realities of conflict, displacement, and chronic disease. Future efforts must prioritise evidence-informed, scalable interventions that bridge humanitarian and health system divides, ensuring that displaced populations are not left behind in global palliative care agendas.

## Supporting information

**S1 Fig. Location of study sites.** Base map data © OpenStreetMap contributors (ODbL). Map annotations by the authors. (DOCX)

## Acknowledgments

We are deeply grateful to all participants who generously shared their time and experiences. Their openness and trust made this study possible, providing novel insights to guide equitable and compassionate care for displaced populations. We would also like to sincerely thank the field researchers and data collectors working in Nyumanzi, Bidi Bidi and Kyaka settlements for their dedicated efforts in recruiting participants and conducting interviews with sensitivity and respect. Their commitment and expertise were invaluable in facilitating meaningful engagement and ensuring the voices of participants were accurately represented. Finally, we acknowledge the support of local health workers and partner organisations in facilitating access and building community trust to support the delivery of the project.

## Author contributions

**Conceptualization:** Matthew J. Allsop.

**Data curation:** Ning Song, Agatha Aduro-Agema, Matthew J. Allsop.

**Formal analysis:** Ning Song, Agatha Aduro-Agema, Eve Namisango, Matthew J. Allsop.

**Funding acquisition:** Eve Namisango, Matthew J. Allsop.

**Investigation:** Agatha Aduro-Agema, Eve Namisango, Vicky Opia, Raphael Rabonye, Emmanuel Mutoto, Matthew J. Allsop.

**Methodology:** Agatha Aduro-Agema, Eve Namisango, Mhoira Leng, Bassey Ebenso, Shaunna Burke, Raphael Rabonye, Matthew J. Allsop.

**Project administration:** Ning Song, Agatha Aduro-Agema, Eve Namisango, Matthew J. Allsop.

**Supervision:** Matthew J. Allsop.

**Validation:** Ning Song, Agatha Aduro-Agema, Eve Namisango, Vicky Opia, Raphael Rabonye, Emmanuel Mutoto, Matthew J. Allsop.

**Visualization:** Ning Song, Matthew J. Allsop.

**Writing – original draft:** Ning Song, Agatha Aduro-Agema, Matthew J. Allsop.

**Writing – review & editing:** Ning Song, Agatha Aduro-Agema, Eve Namisango, Sahla Aroussi, Ping Guo, Mhoira Leng, Bassey Ebenso, Shaunna Burke, Vicky Opia, Raphael Rabonye, Emmanuel Mutoto, Felix Muehlensiepen, Hanna Kaade, William E. Rosa, Eme Asuquo, Richard A Powell, Matthew J. Allsop.

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
