## [Decision Letter · Decision Letter 0]

21 Oct 2025

PGPH-D-25-02762

“…I feel like I am just staying here waiting for death”: A qualitative study of the lived experiences of people with advanced illness in refugee settlements in Uganda.

Dear Dr. Allsop,

Thank you for submitting your manuscript to PLOS Global Public Health. After careful consideration, we feel that it has merit but does not fully meet PLOS Global Public Health’s publication criteria as it currently stands. Therefore, we invite you to submit a revised version of the manuscript that addresses the points raised during the review process.

**PGPH-D-25-02762_editor_comments**

Can you please explain why there are 17 author on a manuscript with 44 participants? Please provide a detailed description of what every author has contributed and please ensure that you review IJME author guidelines.

**PGPH-D-25-02762_reviewer_comments**

**Reviewer 1: **

**Summary of the research and overall impression**

The article is looking into refugees’ access to health services, zooming on individuals with severe illnesses, in need of palliative care. The authors analyze the status of those people’s access to health services, through the lens of their experience living in camps in Uganda. The article argues that patients with advanced illnesses face structural violence from political and social systems, which syndemic dynamic accentuates preventable suffering.

Considering the points of discussion detailed below, I suggest for the authors to review their arguments and calculations before re-submitting the article to PLOS Global Public Health, to be considered for a second round of review. Even though the case study of palliative care, isn’t discussed enough in the context of a refugee camps, I recommend a deeper review of the theoretical frameworks on the 3D Nexus and Camp Management, as well as a stronger argument towards patients with advanced illnesses. This will strengthen the article on the current gap in the literature and the exploration of paths, building a response to palliative care in fragile contexts.

**Discussion for specific areas of improvement**

There are 3 overall issues I recommend to review:

My initial point concerns the services under the umbrella of palliative care. Given the severity of the patients’ health conditions at such medical stage, a safe and efficient pharmaceutical follow up as well as regular consultations, are of outmost importance. In that regard, a fragmentation of services and their frequency, could critically impair patients’ conditions, and quickly escalate the degradation of their health. If this conclusion is detailed in the article, I suggest to make a **tighter link between the interviewees’ health perception, and the actual quality of care they are receiving – describing the type and frequency of the services they have had inside and outside the camp** .

A second point of attention concerns the aim of humanitarian actors with regards to health services operating in fragile contexts. Initially, a refugee camp is a situation which responds to an emergency, not a planned system agreed upon. In that context, the initial objective aims at addressing the emergencies which abide by the method of triage. People are seen by medical providers, according to the patients’ flow of the moment and the urgency of their health conditions. As a result, the “severe illnesses” criteria could be weighted down in front of, more severe cases. As such, I suggest to make a **stronger argument to how triage affects health access to patients with severe illnesses, and how this differs from the structural violence that all refugees face** .

A final impression touches upon the status of a refugee camp. As mentioned, camps aren’t structured and managed with a definite settlement perspective in mind. Advocacy campaigns targeting local authorities, are often carried out to integrate refugees to the host communities via, for example, public health policies. However, settlements are too often and for a long-period of time, operating “off the grid” by the international community, creating a parallel health system. The unfortunate downside of emergency responses in camps, is that they last too long, and fragment of the continuum of care. That being said, there is a great amount of literature on the limit to what humanitarian operators can do within the boundaries of a refugee camp, and the bridges they build with the host country health system. I suggest a **deeper focus on the 3D nexus, and the strategy within which aid actors (humanitarians, stabilization, and development) are currently operating** .

There are 4 major issues I recommend to review:

An element to be explained is the timing between the different data collection periods. Line 196 to 199: the first set is between July and November 2022, then in April 2024 – almost a year and 5 months after, then between August and November 2024.

Considering the 2022-2024 gap, how did you stabilize external factors which could have biased your study? (Like a change in Public Health Policies, Change in international fundings, etc....)

An element I recommend to strengthen, concerns the interviewers and their influence on the type and quality of the data. Line 172 (health workers); line 199-201 (2 researchers); line 202 (clinical officer and a psychologist; then a social anthropologist and a social worker); line 205-2026 (2 nurses, then a nurse and a social worker).

How did you train the data collectors to harmonize the data collection?

In addition, if the article briefly mentions the interviewers’ background as a limit, I recommend to be specific on how the authors addressed this, at different steps of the study: specifically, when the data were collected via the semi-structured interviews, and during the back and forth for the coding and analysis. For those 2 times, during which interviewers worked with coders to complete the translation (line 212):

How did you control or factor-in, the influence of the interviewers on the coder and the analyst, especially considering the 5 languages of the study?

Did the independent authors spoke both the 5 languages to ensure the triangulation (line 221-225)?

The calculation included in the Table 2 pages 13 and 14, needs to be reviewed.

The percentage for the variables, sex, diagnosis, performance, education, marital status, caregiver presence, don’t seem to equal 100%. (For example: 27 women interviewed out of 44 participants is 61,3%, not 56,25% as mentioned line 247 and in the table)

Diagnosis – Cancer (unknown type): the only the data entered is “1”, in the table for Bidibidi but the total is recorded as 2. The total of participants for this variable becomes 45 instead of the initial 44.

There are 2 minor issues I recommend to review:

The initial point concerns the participants, for the pre-test and the study itself:

Line 194, the number of participants should be precise. Is it 2 or 3? I suggest to be specific. I would also recommend adding information on how those 2 or 3 participants were identified to strengthen the methodology.

The elements described in lines 743-745, are part of the essential health care package of services provided by local and international organizations. There are included in the CHS and Sphere standards, and should be a standard practice.

I recommend to clarify the source of your argument. I also suggest adding a published reference and/or specific information on the context.

**Reviewer 2: **

Thank you for the opportunity to read this manuscript, which examines the experiences of (refugees) people living with advanced illness in Uganda. The study is both important and timely. In many conflict-affected countries, populations are ageing and the incidence of advanced disease is increasing alongside rising life expectancy. As a result, this issue will become an even more critical theme in the coming years, particularly in humanitarian and low-resource settings where health systems are already overstretched. The article is carefully executed and offers significant added value to the field of global health research, as it highlights multiple barriers and traumatic experiences faced by individuals in need of palliative care, as well as those living as refugees.

The research question is clearly expressed and well justified. The chosen framework (a phenomenological paradigm) is highly appropriate for a study that seeks to explore personal experiences and subjective meanings of lived reality. The thematic analysis is well structured and thoughtfully divided into suitable subthemes. The interviews successfully bring out the authentic voices of the participants. This indicates a strong level of trust between interviewers and respondents, as the material includes highly sensitive, painful, and even taboo experiences that individuals would not easily disclose without a sense of safety and rapport. The results are presented clearly and coherently.

Minor suggestions

1) The reader might benefit from a simple map of Uganda indicating the locations of the settlements where interviews were conducted, as this would provide contextual understanding of the study sites

2) The ethics statement is written twice in the manuscript (lines 120-122 and 238-241) Remove duplication

I recommend the manuscript for publication with minor revisions. This important work will make a valuable contribution to the literature on palliative care in humanitarian settings. Thank you.

A rebuttal letter that responds to each point raised by the editor and reviewer(s). You should upload this letter as a separate file labeled 'Response to Reviewers'.

We look forward to receiving your revised manuscript.

Kind regards,

Baldeep Kaur Dhaliwal, PhD

Academic Editor

Journal Requirements:

1. Please send a completed 'Competing Interests' statement, including any COIs declared by your co-authors. If you have no competing interests to declare, please state "The authors have declared that no competing interests exist". Otherwise please declare all competing interests beginning with the statement "I have read the journal's policy and the authors of this manuscript have the following competing interests:"

2. Your current Financial Disclosure states, “This research was undertaken with support from the University of Leeds GCRF and Newton Consolidation Accounts (GNCA) funding scheme. WER is partially supported by NIH/NCI comprehensive cancer center award P30 CA008748 and the Robert Wood Johnson Foundation Harold Amos Medical Faculty Development Program. RAP was supported by the National Institute for Health and Care Research (NIHR) Applied Research Collaboration Northwest London. The views expressed by RAP are not necessarily those of the NIHR or the Department of Health and Social Care, London, UK. The funders had no involvement in the design, conduct or development of this research manuscript.”. However, your funding information on the submission form indicates that you received funding from “University of Leeds GCRF and Newton Consolidation Accounts”. Please indicate by return email the full and correct funding information for your study and confirm the order in which funding contributions should appear. Please be sure to indicate whether the funders played any role in the study design, data collection and analysis, decision to publish, or preparation of the manuscript.

3. In the online submission form, you indicated that The data that support the findings of this study are available from the corresponding author, upon reasonable request.

3. Uploaded as supplementary information.

4. Please provide separate figure files in .tif or .eps format.

Reviewers' comments:

Reviewer's Responses to Questions

**Comments to the Author**

1. Does this manuscript meet PLOS Global Public Health’s publication criteria?

Reviewer #1: Partly

Reviewer #2: Yes

2. Has the statistical analysis been performed appropriately and rigorously?

Reviewer #1: No

Reviewer #2: Yes

3. Have the authors made all data underlying the findings in their manuscript fully available (please refer to the Data Availability Statement at the start of the manuscript PDF file)?

Reviewer #1: Yes

Reviewer #2: Yes

4. Is the manuscript presented in an intelligible fashion and written in standard English?

Reviewer #1: Yes

Reviewer #2: Yes

Reviewer #1: PGPH-D-25-02762_reviewer_comments

Summary of the research and overall impression

The article is looking into refugees’ access to health services, zooming on individuals with severe illnesses, in need of palliative care. The authors analyze the status of those people’s access to health services, through the lens of their experience living in camps in Uganda. The article argues that patients with advanced illnesses face structural violence from political and social systems, which syndemic dynamic accentuates preventable suffering.

Considering the points of discussion detailed below, I suggest for the authors to review their arguments and calculations before re-submitting the article to PLOS Global Public Health, to be considered for a second round of review. Even though the case study of palliative care, isn’t discussed enough in the context of a refugee camps, I recommend a deeper review of the theoretical frameworks on the 3D Nexus and Camp Management, as well as a stronger argument towards patients with advanced illnesses. This will strengthen the article on the current gap in the literature and the exploration of paths, building a response to palliative care in fragile contexts.

Discussion for specific areas of improvement

There are 3 overall issues I recommend to review:

My initial point concerns the services under the umbrella of palliative care. Given the severity of the patients’ health conditions at such medical stage, a safe and efficient pharmaceutical follow up as well as regular consultations, are of outmost importance. In that regard, a fragmentation of services and their frequency, could critically impair patients’ conditions, and quickly escalate the degradation of their health. If this conclusion is detailed in the article, I suggest to make a tighter link between the interviewees’ health perception, and the actual quality of care they are receiving – describing the type and frequency of the services they have had inside and outside the camp.

A second point of attention concerns the aim of humanitarian actors with regards to health services operating in fragile contexts. Initially, a refugee camp is a situation which responds to an emergency, not a planned system agreed upon. In that context, the initial objective aims at addressing the emergencies which abide by the method of triage. People are seen by medical providers, according to the patients’ flow of the moment and the urgency of their health conditions. As a result, the “severe illnesses” criteria could be weighted down in front of, more severe cases. As such, I suggest to make a stronger argument to how triage affects health access to patients with severe illnesses, and how this differs from the structural violence that all refugees face.

A final impression touches upon the status of a refugee camp. As mentioned, camps aren’t structured and managed with a definite settlement perspective in mind. Advocacy campaigns targeting local authorities, are often carried out to integrate refugees to the host communities via, for example, public health policies. However, settlements are too often and for a long-period of time, operating “off the grid” by the international community, creating a parallel health system. The unfortunate downside of emergency responses in camps, is that they last too long, and fragment of the continuum of care. That being said, there is a great amount of literature on the limit to what humanitarian operators can do within the boundaries of a refugee camp, and the bridges they build with the host country health system. I suggest a deeper focus on the 3D nexus, and the strategy within which aid actors (humanitarians, stabilization, and development) are currently operating.

There are 4 major issues I recommend to review:

An element to be explained is the timing between the different data collection periods. Line 196 to 199: the first set is between July and November 2022, then in April 2024 – almost a year and 5 months after, then between August and November 2024.

• Considering the 2022-2024 gap, how did you stabilize external factors which could have biased your study? (Like a change in Public Health Policies, Change in international fundings, etc....)

An element I recommend to strengthen, concerns the interviewers and their influence on the type and quality of the data. Line 172 (health workers); line 199-201 (2 researchers); line 202 (clinical officer and a psychologist; then a social anthropologist and a social worker); line 205-2026 (2 nurses, then a nurse and a social worker).

• How did you train the data collectors to harmonize the data collection?

In addition, if the article briefly mentions the interviewers’ background as a limit, I recommend to be specific on how the authors addressed this, at different steps of the study: specifically, when the data were collected via the semi-structured interviews, and during the back and forth for the coding and analysis. For those 2 times, during which interviewers worked with coders to complete the translation (line 212):

• How did you control or factor-in, the influence of the interviewers on the coder and the analyst, especially considering the 5 languages of the study?

• Did the independent authors spoke both the 5 languages to ensure the triangulation (line 221-225)?

The calculation included in the Table 2 pages 13 and 14, needs to be reviewed.

• The percentage for the variables, sex, diagnosis, performance, education, marital status, caregiver presence, don’t seem to equal 100%. (For example: 27 women interviewed out of 44 participants is 61,3%, not 56,25% as mentioned line 247 and in the table)

• Diagnosis – Cancer (unknown type): the only the data entered is “1”, in the table for Bidibidi but the total is recorded as 2. The total of participants for this variable becomes 45 instead of the initial 44.

There are 2 minor issues I recommend to review:

The initial point concerns the participants, for the pre-test and the study itself:

• Line 194, the number of participants should be precise. Is it 2 or 3? I suggest to be specific. I would also recommend adding information on how those 2 or 3 participants were identified to strengthen the methodology.

The elements described in lines 743-745, are part of the essential health care package of services provided by local and international organizations. There are included in the CHS and Sphere standards, and should be a standard practice.

• I recommend to clarify the source of your argument. I also suggest adding a published reference and/or specific information on the context.

**Reviewer #2: ** Thank you for the opportunity to read this manuscript, which examines the experiences of (refugees) people living with advanced illness in Uganda. The study is both important and timely. In many conflict-affected countries, populations are ageing and the incidence of advanced disease is increasing alongside rising life expectancy. As a result, this issue will become an even more critical theme in the coming years, particularly in humanitarian and low-resource settings where health systems are already overstretched. The article is carefully executed and offers significant added value to the field of global health research, as it highlights multiple barriers and traumatic experiences faced by individuals in need of palliative care, as well as those living as refugees.

The research question is clearly expressed and well justified. The chosen framework (a phenomenological paradigm) is highly appropriate for a study that seeks to explore personal experiences and subjective meanings of lived reality. The thematic analysis is well structured and thoughtfully divided into suitable subthemes. The interviews successfully bring out the authentic voices of the participants. This indicates a strong level of trust between interviewers and respondents, as the material includes highly sensitive, painful, and even taboo experiences that individuals would not easily disclose without a sense of safety and rapport. The results are presented clearly and coherently.

Minor suggestions

1) The reader might benefit from a simple map of Uganda indicating the locations of the settlements where interviews were conducted, as this would provide contextual understanding of the study sites

2) The ethics statement is written twice in the manuscript (lines 120-122 and 238-241) Remove duplication

I recommend the manuscript for publication with minor revisions. This important work will make a valuable contribution to the literature on palliative care in humanitarian settings. Thank you.

**Do you want your identity to be public for this peer review?** For information about this choice, including consent withdrawal, please see our Privacy Policy

Reviewer #1: No

Reviewer #2: **Yes: ** Agneta Kallström

---

## [Decision Letter · Decision Letter 1]

3 Dec 2025

“…I feel like I am just staying here waiting for death”: A qualitative study of the lived experiences of people with advanced illness in refugee settlements in Uganda.

PGPH-D-25-02762R1

Dear Allsop,

We are pleased to inform you that your manuscript '“…I feel like I am just staying here waiting for death”: A qualitative study of the lived experiences of people with advanced illness in refugee settlements in Uganda.' has been provisionally accepted for publication in PLOS Global Public Health.

Best regards,

Baldeep Kaur Dhaliwal, PhD

Academic Editor

Reviewer Comments (if any, and for reference):

Reviewer's Responses to Questions

**Comments to the Author**

Reviewer #1: All comments have been addressed

Reviewer #2: All comments have been addressed

publication criteria?

Reviewer #1: Yes

Reviewer #2: Yes

3. Has the statistical analysis been performed appropriately and rigorously?

Reviewer #1: Yes

Reviewer #2: N/A

4. Have the authors made all data underlying the findings in their manuscript fully available (please refer to the Data Availability Statement at the start of the manuscript PDF file)?

Reviewer #1: Yes

Reviewer #2: Yes

5. Is the manuscript presented in an intelligible fashion and written in standard English?

Reviewer #1: Yes

Reviewer #2: Yes

Reviewer #1: PGPH-D-25-02762_reviewer_comments

Overall impression of the revised manuscript

The methodology was reviewed and consolidated, addressing the calculation errors, the data collection risks, and the potential coding bias.

In addition, the manuscript was integrated into a larger, technical (medical/health system), operational (3D nexus), and political (national authorities) discussion, alerting the international community on underrepresented patients in fragile settings.

I thank the authors for integrating a tighter theoretical framework, and a deeper analytical section. This research will surely contribute to, improving field operations, and ultimately patients’ care.

I recommend the manuscript for publication.

Reviewer #2: No

**Do you want your identity to be public for this peer review?** For information about this choice, including consent withdrawal, please see our Privacy Policy

Reviewer #1: No

Reviewer #2: **Yes: ** Agneta Kallström
